# Anaplastic Lymphoma Kinase- and CD30-Positive Anaplastic Large-Cell Lymphoma of the External Auditory Canal

**DOI:** 10.3390/diagnostics11071220

**Published:** 2021-07-06

**Authors:** Shih-Lung Chen, Kai-Chieh Chan

**Affiliations:** 1Department of Otolaryngology & Head and Neck Surgery, Chang Gung Memorial Hospital, Linkou 333, Taiwan; rlong289@gmail.com; 2School of Medicine, Chang Gung University, Taoyuan 333, Taiwan

**Keywords:** anaplastic large-cell lymphoma, anaplastic lymphoma kinase, otology, head and neck, external auditory canal

## Abstract

Anaplastic large-cell lymphoma (ALCL), a form of non-Hodgkin’s lymphoma, is rare in the external auditory canal (EAC). ALCL in the EAC region is challenging for clinicians and pathologists. We report a 34-year-old male presented with the chief complaint of a painless mass in the left EAC for approximately 2 weeks. Anaplastic lymphoma kinase (ALK)- and CD30-positive ALCL were confirmed using computed tomography, positron emission tomography, histopathological examination and IHC staining. We compared the clinicopathological characteristics of our patient with those of previous cases. The biopsy and IHC findings confirmed the rare diagnosis of ALK- and CD30-positive ALCL of the EAC. Radiotherapy and concurrent chemoradiotherapy are indicated for lymphoma depending on the extent of the disease. Brentuximab vedotin as initial salvage therapy should be considered for recurrent or refractory ALK-positive ALCL.

## 1. Introduction

Anaplastic large-cell lymphoma (ALCL) was first described by Stein et al. in 1985 as a pleomorphic cell malignancy with expression of the CD30-positive marker [1]. The gold standard for the diagnosis of CD30-positive ALCL is positive staining for CD30 in at least 75% of the lymphoma cells [2]. In addition, a comprehensive histopathological analysis, including immunohistochemical (IHC) staining for anaplastic lymphoma kinase (ALK), is needed to establish the diagnosis because the management and outcomes differ according to the immunophenotype [3].

ALCL of the external auditory canal (EAC) is an extremely rare condition; three cases of CD30-positive ALCL have been reported in the literature, none of which showed ALK-positive cells [4,5,6]. Herein, we describe, for the first time, CD30-positive and ALK-positive ALCL of the EAC in a 34-year-old male.

## 2. Case Presentation

A 34-year-old male without systemic disease presented to our otorhinolaryngology department with a painless mass in the left ear for 2 weeks. A physical examination revealed a protruding tumor with bleeding tendency and otorrhea and crusting leading to total occlusion of the EAC (Figure 1). No vertigo, facial palsy, or head or neck field lesions were observed.

The laboratory findings revealed a white blood cell count of 9000/µL (normal: 3900–10,600/µL), a platelet count of 320,000/µL (normal: 150,000–400,000/µL) and a lymphocyte percent of 15.1% (normal range 20–56%).

Computed tomography (CT) with contrast revealed an intensely enhancing soft tissue mass approximately 3.5 cm in diameter in the left EAC extending to the adjacent subcutaneous area (Figure 2).

A tumor biopsy was performed. The pathologic sections showed the skin with diffuse infiltration of large tumor cells with a mitotic pattern (Figure 3A). IHC analysis revealed diffuse lymphoid infiltration with strong positive staining for CD3, CD30 and ALK but negative for CD20 and CD56 (Figure 3B–D). A diagnosis of CD30-positive and ALK-positive anaplastic cutaneous primary ALCL of the EAC was confirmed. Positron emission tomography revealed radiotracer uptake in the left EAC (Figure 4). 

The patient underwent a course of concurrent radiochemotherapy (CCRT) with 30 Gy radiotherapy and 6 cycles of cyclophosphamide, doxorubicin, vincristine and prednisone (CHOP); however, the tumor recurred 6 months later. The patient then underwent four cycles of Brentuximab vedotin and has been followed by a hematologist for 1 year without recurrence.

## 3. Discussion

Lymphomas account for 2.5% of head and neck malignancies [7]. ALCL of the EAC is a primary cutaneous T-cell lymphoma, and a rare form of non-Hodgkin’s lymphoma [6]. It occurs primarily in adults aged between 40 and 60 years of age and rarely affects pediatric and adolescent populations [3,6].

ALCL of the EAC has been associated with trauma, immunosuppressive states and the Epstein–Barr virus [3]; however, our patient had no medical or systemic disease history.

Primary cutaneous CD30-positive T-cell lymphoma typically presents as a single or multiple reddish tumors [6]. In our patient, the tumor manifested as a painless mass accompanied by otorrhea and crusting.

A comprehensive tumor survey is recommended; however, because the tumor in our case presented as a nonspecific cutaneous lesion, a biopsy was essential for an accurate diagnosis [6]. Indeed, malignant EAC tumors can be easily misdiagnosed. Malignant tumors often present as treatment-resistant chronic otitis [4]. Marcal et al. described a patient with an initial diagnosis of cerumen accumulation [6], and Zhang et al. concluded that EAC tumors are frequently misdiagnosed as otitis externa or otitis media, leading to delayed treatment [8].

Pathological sections often show a diffuse infiltration of lymphoid cells [5] and diverse activated T-phenotypes (CD2+, CD3+, CD4+, CD45RO+, CD25+, CD30+) [3]. The gold standard for ALCL diagnosis is CD30-positive staining in at least 75% of the large lymphoma cells infiltrating the dermis and hypodermis [2]. For this patient, the different diagnosis should include extranodal NK/T cell lymphoma because CD30 will also be expressed in NK/T cell lymphoma. NK/T cell lymphoma is positive for CD56 and EBER [9], but ALCL is negative for these two stainings. The difference in staining could be utilized to differentiate these two malignancies.

Previously reported cases of ALCL of the EAC are shown in Table 1. The prognosis differs in ALK-positive and ALK-negative ALCL [3]. ALK-positivity was not presented in the three reported cases of ALCL of the EAC. Rassidakis et al. noted that patients with ALK-positive ALCL have a more favorable prognosis than those with ALK-negative tumors [10]; however, the reason for this difference is not clear.

The treatment options for ALCL of the EAC are based on the extent of involvement [3]. As primary cutaneous CD30-positive T-cell lymphoma is a radiosensitive tumor, radiotherapy (RT) is indicated when complete surgical removal is not possible [6,11]. Chemotherapy with CHOP is considered for patients with extensive skin involvement or visceral spread [12]. In our patient, the initial assessment showed a diffuse infiltration of large tumor cells in a mitotic pattern; therefore, CCRT was the first-line treatment.

Prognosis is favorable in most patients with a reported 10-year survival rate of more than 90% [3,13]. Regional lymph node involvement is rare and positive lymph nodes do not appear to adversely affect the prognosis [6]. In fact, primary cutaneous ALCL has a better prognosis and more indolent course than systemic ALCL does [14]. Furthermore, the treatment and outcomes differ according to the positive or negative expression of the ALK marker. ALK-positive ALCL is diagnosed at a younger age and has a better outcome with lower incidences of recurrence and less progression to systemic disease [14].

After completing the CCRT course, our patient had local cutaneous recurrence. Cutaneous recurrence is not uncommon with up to 40% of treated patients experiencing recurrence. Thus, long-term follow-up is recommended. Moreover, 12–16% of patients progress to systemic disease [14].

In fact, radiotherapy is mainly applied because CD30-positive ALCL is an indolent and radiosensitive tumor, especially when complete surgical resection is not possible [6,11]. CCRT is primarily reserved for the patients with diffuse skin spread or advanced anatomical involvement [12]. In patients with recurrent or refractory ALCL, brentuximab vedotin, an antibody-drug conjugate, should be considered as initial salvage therapy for ALK-positive and ALK-negative ALCL [15]. A previous study found that patients with intractable ALCL who achieved a complete response with brentuximab vedotin had 79% overall survival and 57% progression-free survival at 5 years [16]. The most common side effects of brentuximab vedotin were fatigue, pyrexia, diarrhea, nausea, neutropenia and peripheral neuropathy [17].

Currently, a trial displayed that the effect of brentuximab vedotin, cyclophosphamide, doxorubicin and prednisone (A + CHP) is superior to that of CHOP for the treatment of CD30-positive peripheral T-cell lymphomas in progression-free survival and overall survival. The median progression-free survival of A + CHP group and CHOP group were 48.2 months and 20.8 months, respectively [18].

## 4. Conclusions

We describe a patient who presented with a painless mass in the EAC accompanied by otorrhea and crusting. The biopsy and IHC findings confirmed the diagnosis of ALK-positive and CD30-positive ALCL of the EAC. RT and CCRT were indicated based on the extent of disease. Brentuximab vedotin should be considered an initial salvage therapy in patients with recurrent or intractable ALK-positive ALCL.

## Figures and Tables

**Figure 1 diagnostics-11-01220-f001:**
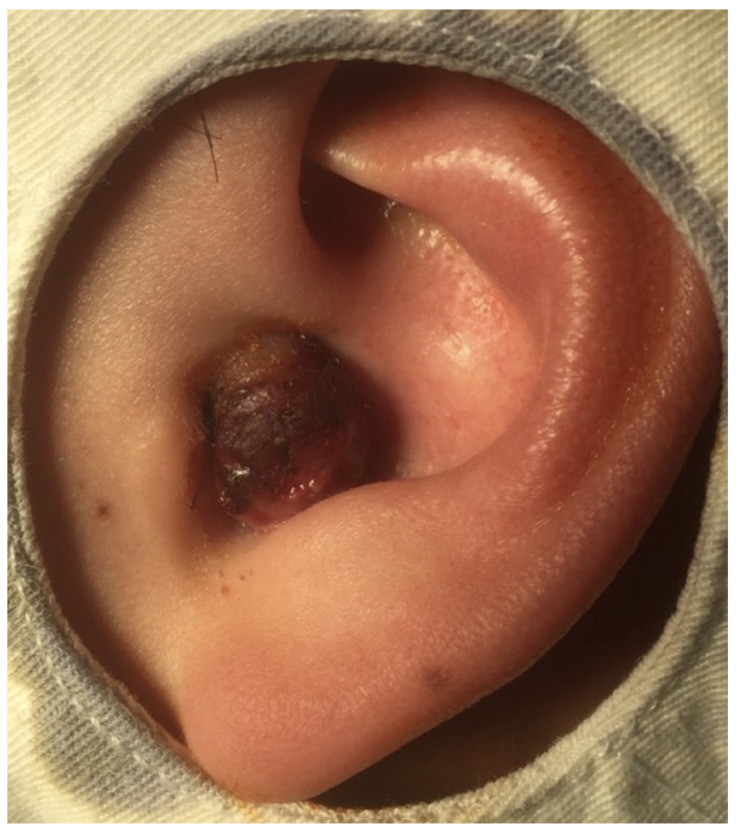
An exophytic and protruding mass with crusting and touching bleeding, leading to total occlusion of the external auditory canal.

**Figure 2 diagnostics-11-01220-f002:**
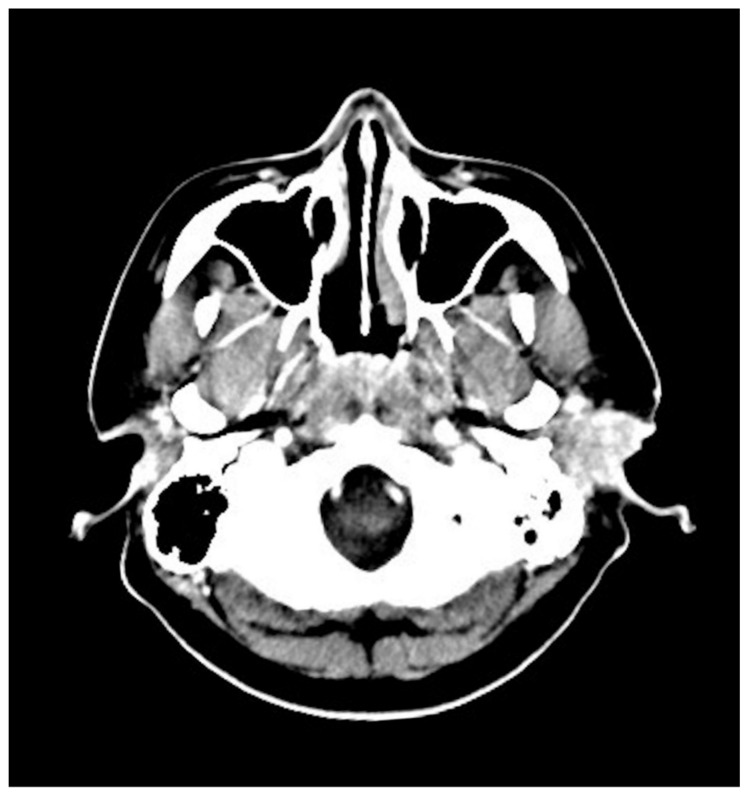
Computed tomography scan showing a heterogeneous enhancing soft tissue mass approximately 3.5 cm in diameter in the left EAC without obvious osseous erosion.

**Figure 3 diagnostics-11-01220-f003:**
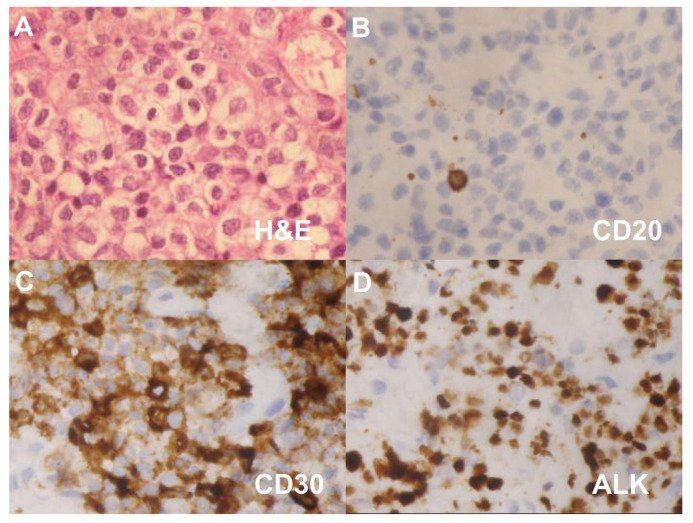
(**A**) Pathological sections showing diffuse infiltration of large tumor cells with mitotic figures. Immunohistochemical findings showing (**B**) CD20-negative, (**C**) CD30-positive and (**D**) ALK-positive tumor cells. (Original magnification × 400 for **A**–**D**).

**Figure 4 diagnostics-11-01220-f004:**
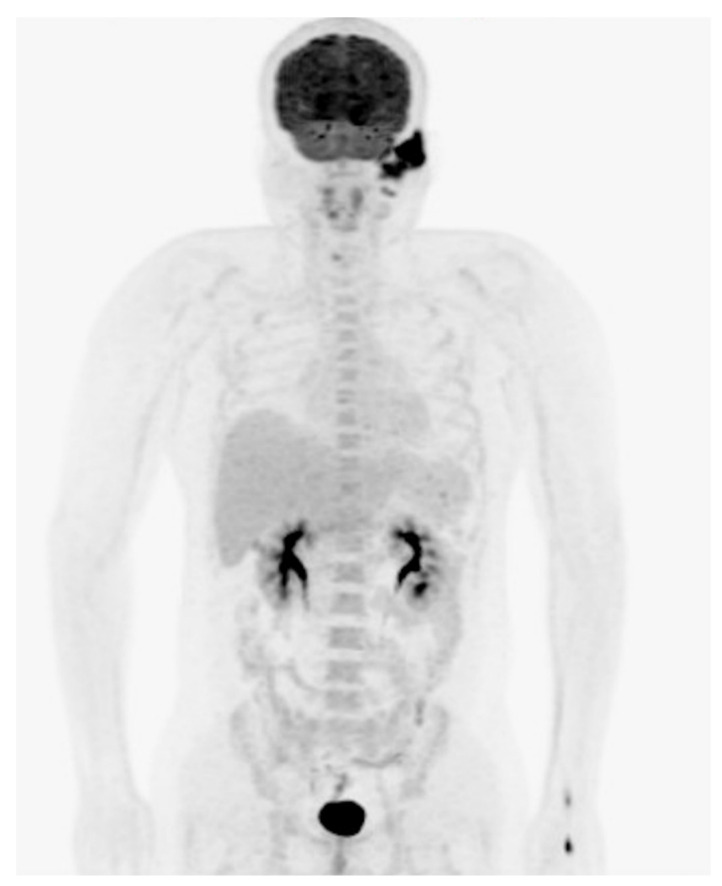
Positron emission tomography images showing radiotracer uptake in the left EAC.

**Table 1 diagnostics-11-01220-t001:** Reported cases of anaplastic large-cell lymphoma of the external auditory canal.

Authors (Year)	Gender	Age (Y)	Initial Symptoms	CT	PET	IHC Positive	ALK	Treatment
Merkus et al., 2000	Female	83	Otalgia, otorrhea	Bony destruction of EAC	NA	CD30	NA	RT
Gomaa et al., 2010	Female	47	Otalgia, EAC swelling	No osseous erosion	Radio-uptake	CD4, CD5, CD30	NA	RT
Marcal et al., 2012	Female	68	Otalgia, hypoacusis	No osseous erosion	NA	CD3, CD5, CD30	Negative	RT
Present case, 2021	Male	34	Painless mass	No osseous erosion	Radio-uptake	CD3, CD30	Positive	CCRT

Y: years; CT: computed tomography; PET: positron emission tomography; IHC: immunohistochemistry; AKL: Anaplastic lymphoma kinase; RT: radiotherapy; CCRT: concurrent chemoradiotherapy.

## Data Availability

All data generated or analyzed during this study are included in this published article.

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
