# Peer review of "Anaplastic Lymphoma Kinase- and CD30-Positive Anaplastic Large-Cell Lymphoma of the External Auditory Canal"

_diagnostics, 2021, doi:10.3390/diagnostics11071220_

Round 1

Reviewer 1 Report

The authors mentioned CD30+ALK+ cALCL of the EAC for the first time. This case was treated by CHOP with radiation, followed by Brentuximab vedotin. However, this paper should be improved.

  1. HE picture is quite poor image. You should take a picture with higher quality and x400 magnification. Atypical cells are unclear.
  2. Are figure3 C,D also x100 magnification? These don’t look same magnification as figure3A.
  3. You should add total Gy of RT.
  4. How many cycles of Brentuximab vedotin did the patient receive?
  5. Did you rule out a possibility of NK/T cell lymphoma? CD30 can be expressed in NK/T cell lymphoma. CD56 and EBER staining are needed.
  6. The authors should mention the recommended orders of the treatment (RT, CCRT, brentuximab vedotin), more clearly. Did you mention about the comparison of the effect of CHOP with brentuximab vedotin?

Author Response

Revision of “Anaplastic lymphoma kinase- and CD30-positive anaplastic large-cell lymphoma of the external auditory canal”

Dear Editor and reviewers,

Thank you for the thoughtful and detailed review. In the following pages, we respond in a point-by-point manner to all comments of the reviewers. In the revision, all changes are highlighted in RED color. During this valuable process of revision to enhance the manuscript, we really thank the editors and reviewers for their insightful and informative comments. Please let us know if additional clarification is required. We deeply appreciate your valuable comments, which encouraged us to consider our work more thoroughly. Thank you again.

Yours sincerely,

Corresponding author, Kai-Chieh Chan, M.D.

Department of Otolaryngology & Head and Neck Surgery, Chang Gung Memorial Hospital, Linkou, Taiwan

E-mail: kjchan5109@gmail.com

Tel: +886-2-24313131 ext 2445; +886-2-24313161

Reviewer 1

The authors mentioned CD30+ALK+ cALCL of the EAC for the first time. This case was treated by CHOP with radiation, followed by Brentuximab vedotin. However, this paper should be improved.

Comment 1: HE picture is quite poor image. You should take a picture with higher quality and x400 magnification. Atypical cells are unclear.

Reply 1:

Yes, we appreciate the reviewer’s insightful comments. We completely agreed.

We revised the quality and magnification of Figure 3A. We corrected the magnification to x400 magnification and uploaded the new Figure 3A on Page 3 line 62. In addition, we also selected areas with clear atypical cells.

Comment 2: Are Figure3 C, D also x100 magnification? These don’t look same magnification as Figure3A.

Reply 2: Yes, we thank for reviewer’ valuable comments.

Indeed, the original Figure 3C & D were different from Figure 3A in magnification. Thank for reviewer’s detailed review.

We revised all four pictures of Figure 3A, B , C and D to version of x400 magnification, and re-uploaded the new Figure 3A-D in Page 3 line 62. We also added “(Original magnification, x 400 for A-D)” on Page 3 lines 64-65.

Comment 3: You should add total Gy of RT.

Reply 3:Yes, we thank for the reviewer’s comment. We totally appreciate.

This patient received the 30Gy radiotherapy and 6 cycles of CHOP treatment clinically. We revised the original text to "The patient underwent a course of concurrent radiochemotherapy (CCRT) with 30Gy radiotherapy and 6 cycles of cyclophosphamide, doxorubicin, vincristine and prednisone (CHOP)" on Page 2 lines 51-53.

Comment 4: How many cycles of Brentuximab vedotin did the patient receive?

Reply 4:

Yes, we thank for reviewer’ comment. We totally agree.

This patient received 4 cycles of Brentuximab vedotin clinically. We modified the original text to "The patient then underwent 4 cycles of Brentuximab vedotin" on Page 2 lines 53-54.

Comment 5: Did you rule out a possibility of NK/T cell lymphoma? CD30 can be expressed in NK/T cell lymphoma. CD56 and EBER staining are needed.

Reply 5:

Yes, we appreciate the reviewer’s insightful comments. We completely agreed.

This patient’s CD56 was negative clinically. We added "IHC analysis revealed diffuse lymphoid infiltration with strong positive staining for CD3, CD30, ALK but negative for CD20, CD56" on Page 2 lines 46-48.

For Epstein-Barr virus-expressed RNA (EBER) of NK/T cell lymphoma mentioned by reviewer 1, we did not perform the test in this patient.

However, because it is a valuable opinion, we added the relevant content in the discussion:

"For this patient, the differential diagnosis should include extranodal NK/T cell lymphoma because CD30 will also be expressed in NK/T cell lymphoma. NK/T cell lymphoma is positive for CD56 and EBER [9], but ALCL is negative for these two staining. The difference in staining could be utilized to differentiate these two malignancies " on Page 5 lines 88-92

The new reference 9 are presented as follows:

  1. Wang, W.; Nong, L.; Liang, L.; Zheng, Y.; Li, D.; Li, X.; Li, T. Extranodal NK/T-cell lymphoma, nasal type without evidence of EBV infection. Oncol Lett 2020, 20, 2665-2676, doi:10.3892/ol.2020.11842.

Comment 6: The authors should mention the recommended orders of the treatment (RT, CCRT, brentuximab vedotin), more clearly. Did you mention about the comparison of the effect of CHOP with brentuximab vedotin?

Reply 6: We thank for the reviewer’s comment. We totally appreciate.

We reorganized the recommended orders of the treatment (RT, CCRT, brentuximab vedotin) and revised as the following content:

“In fact, radiotherapy is mainly applied because CD30-positive ALCL is an indolent and radiosensitive tumor, especially when complete surgical resection is not possible [6,11]. CCRT is primarily reserved for the patients with diffuse skin spread or advanced anatomical involvement [12]. In patients with recurrent or refractory ALCL, brentuximab vedotin, an antibody-drug conjugate, should be considered as initial salvage therapy for ALK-positive and ALK-negative ALCL [15].” on Page 5 lines 117-122.

As review 1 said, we didn’t mention about the comparison of the comparison of the effect of CHOP with brentuximab vedotin. We consider it’s an insightful opinion.

However, we did not find a direct comparison of the effect of CHOP with brentuximab vedotin in literature, but we found that there was a trial comparing the effect of brentuximab vedotin, cyclophosphamide, doxorubicin, and prednisone (A+CHP) with that of CHOP, and we added the following sentences to the discussion:

“Currently, a trial displayed that the effect of brentuximab vedotin, cyclophosphamide, doxorubicin, and prednisone (A+CHP) is superior to that of CHOP for the treatment of CD30-positive peripheral T-cell lymphomas in progression-free survival and overall survival. The median progression-free survival of A+CHP group and CHOP group were 48.2 months and 20.8 months, respectively [18].” on Page 5 line 127 to Page 6 line 131.

The new reference 18 are presented as follows:

  1. Horwitz, S.; O'Connor, O.A.; Pro, B.; Illidge, T.; Fanale, M.; Advani, R.; Bartlett, N.L.; Christensen, J.H.; Morschhauser, F.; Domingo-Domenech, E.; et al. Brentuximab vedotin with chemotherapy for CD30-positive peripheral T-cell lymphoma (ECHELON-2): a global, double-blind, randomised, phase 3 trial. Lancet 2019, 393, 229-240, doi:10.1016/S0140-6736(18)32984-2.

Reviewer 2 Report

The authors reported a clinical case of anaplastic large-cell lymphoma (ALCL) in the external auditory canal (EAC), which is a very rare condition. They described the painless mass in the EAC and found it is ALK- and CD30-positive ALCL with biopsy and IHC detections. Furthermore, they treated this patient with Brentuximab vedotin and the disease was not recurrent after 1 year. The references are missing on Page 1, lines 29-31. Therefore, the authors introduced the brief features of ALCL and efficient treatment for the patient, and this is an interesting, enlightening study to report a rare case of ALCL in EAC.

Author Response

Revision of “Anaplastic lymphoma kinase- and CD30-positive anaplastic large-cell lymphoma of the external auditory canal”

Dear Editor and reviewers,

Thank you for the thoughtful and detailed review. In the following pages, we respond in a point-by-point manner to all comments of the reviewers. In the revision, all changes are highlighted in RED color. During this valuable process of revision to enhance the manuscript, we really thank the editors and reviewers for their insightful and informative comments. Please let us know if additional clarification is required. We deeply appreciate your valuable comments, which encouraged us to consider our work more thoroughly. Thank you again.

Yours sincerely,

Corresponding author, Kai-Chieh Chan, M.D.

Department of Otolaryngology & Head and Neck Surgery, Chang Gung Memorial Hospital, Linkou, Taiwan

E-mail: kjchan5109@gmail.com

Tel: +886-2-24313131 ext 2445; +886-2-24313161

Reviewer 2

The authors reported a clinical case of anaplastic large-cell lymphoma (ALCL) in the external auditory canal (EAC), which is a very rare condition. They described the painless mass in the EAC and found it is ALK- and CD30-positive ALCL with biopsy and IHC detections. Furthermore, they treated this patient with Brentuximab vedotin and the disease was not recurrent after 1 year. The references are missing on Page 1, lines 29-31. Therefore, the authors introduced the brief features of ALCL and efficient treatment for the patient, and this is an interesting, enlightening study to report a rare case of ALCL in EAC.

Reply 1:

Yes, we thank for reviewer’ comments. We totally appreciate.

We added 3 cases’ references on Page 1 line 31 [4-6]. We have also renewed the order of our references throughout the article. Thank you for your valuable insights and comments.

Round 2

Reviewer 1 Report

The authors responded for my comments, adequately.